# *IO*SHIFT: BACKDOOR DEFENSE VIA MODEL BIAS SHIFT IN FEDERATED LEARNING

## ABSTRACT

As a privacy-preserving and decentralized machine learning framework, Federated Learning (FL) is vulnerable to backdoor attacks. Current backdoor defenses rely on a strong assumption: defenders have the ability of defining a benign parameter space using gradient information to detect or remove malicious updates. However, in the real-world not-independent-and-identically-distributed (Non-IID) FL scenarios, this is a particularly challenging task, exhibiting inconsistent performance across different systems and settings. In this paper, we reveal the *Backdoor-Induced Model Bias Shift* phenomenon, where the implantation of backdoor shortcuts shifts the model bias on out-of-distribution (OOD) data toward the target class. Inspired by this insight, we propose *IO*Shift, a novel backdoor detection and removal method based on model bias shift in federated learning. *IO*Shift detects malicious updates by measuring bias shifts on OOD data, using the model bias on in-distribution data as a reference. Furthermore, it employs adaptive weight pruning to maintain high utility on clean tasks. *IO*Shift seamlessly integrates into existing FL frameworks without requiring any modifications, such as altering communication protocols or injecting elaborated tasks. Experimental results on benchmark datasets and backdoor attacks demonstrate that *IO*Shift effectively outperforms state-of-the-art backdoor defenses. Code is available here.

## 1 INTRODUCTION

Federated Learning (FL) has been widely applied in various domains, including healthcare, finance, and smart city Shen et al. (2025); Yang et al. (2023b), enabling collaborative model training under the coordination of a center server. In this paradigm, multiple clients trains a shared model locally on their private data and then upload model updates to the FL server which aggregates them to improve the global model iteratively. Notably, FL performs well even in real-world scenarios where client data is not independent and identically distributed (Non-IID) Shi et al. (2025). However, ensuring the integrity of FL remains a significant challenge, particularly in defending against backdoor attacks Naseri et al. (2024); Fan et al. (2025); Li et al. (2025a;b). Adversaries can inject malicious updates during training, embedding backdoor behaviors into the global model while maintaining high performance on clean tasks.

**Limitations.** Recently, many research has focused on defending against backdoor attacks, primarily including backdoor detection and backdoor removal.

*Detection:* Existing detection methods often rely on the assumption that malicious updates exhibit significant differences from benign updates in the parameter space Li & Dai (2024). These methods compute similarity metrics between malicious and benign updates, e.g., cosine similarity or manually defined features. However, recent adversarial backdoor attacks can carefully optimize malicious updates to remain within the benign parameter space Zhang et al. (2023); Lyu et al. (2024). More critically, under Non-IID, the performance of these anomaly detection-based methods significantly degrades Li & Dai (2024). This is because benign updates from different distributions inherently exhibit high dissimilarity, making it nearly impossible to define a clear benign parameter space.

*Removal:* Assuming malicious updates can be accurately detected, the most straightforward removal strategy is to exclude them from aggregation, which inevitably leads to information loss Cao et al. (2023); Blanchard et al. (2017). Another approach is to limit the impact of malicious updates by constraining them within the benign parameter space, such as clipping techniques Nguyen et al.

(2022). However, similar to backdoor detection, defining a benign update parameter space for unknown client is highly challenging in the Non-IID scenarios. Even more absurdly, existing backdoor removal methods impose an unreasonable requirement: cooperation from the attacker—an utterly unacceptable premise Alam et al. (2024); Zhao et al. (2023).

**Main Idea.** In this paper, we propose *IO*Shift, a federated backdoor detection and removal framework based on the model bias shift toward in-distribution (ID) and out-of-distribution (OOD) data. First, we reveal the phenomenon of *Backdoor-Induced Model Bias Shift* for OOD data. As illustrated in Figure 1, a) Model bias arises due to data heterogeneity, leading to imbalanced class accuracy during testing He & Garcia (2009); Wang et al. (2024). b) For unseen data (OOD data), models that obtains the decision boundary based on heterogeneous dataset exhibit a similar model bias Liu et al. (2024a); Ghosh et al. (2024). c) Backdoor embedding biases the model by establishing a stronger activation path (shortcut)

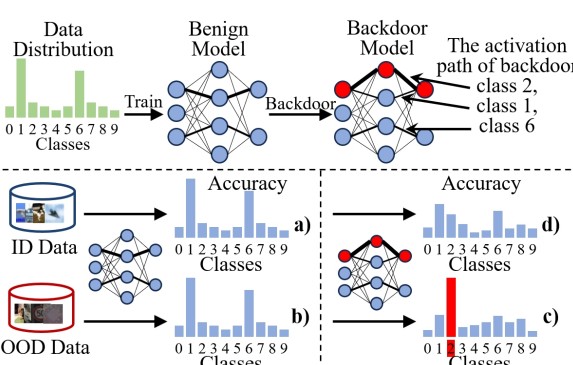

Figure 1: The phenomenon of backdoor-induced model bias shift.

between the trigger and the target class, resulting in an absolute bias toward unknown data. d) Conversely, the activation paths for model bias classes in ID data are significantly suppressed due to the establishment of this strong backdoor shortcut Yang et al. (2023a). See detail in Section 4.1.

Inspired by this, we determine whether a update has been implanted into backdoor by measuring *IO*Shift score between the model bias on ID and OOD data. If the detected score exceeds a threshold (determined based on ASR, see Section 5 for detail), it indicates that the backdoor has affected the model bias, thereby obtaining the target class. To mitigate the backdoor, we employ an adaptive pruning strategy guided by *IO*Shift score to preserve the model's utility on clean tasks. *IO*Shift eliminates the strong assumption of existing defenses that rely on defining a benign parameter space, achieving effective in high Non-IID FL scenarios. Moreover, *IO*Shift can be seamlessly integrated into existing FL frameworks with detection and removal steps without altering the core framework.

**Contributions.** The following are our contributions:

- We reveal the phenomenon of backdoor-induced model bias shift, breaking the limitations of existing benign parameter space defining-based approaches. Since malicious updates are detected individually, *IO*Shift is also applicable to asynchronous FL.

- We propose *IO*Shift, a unified framework for federated backdoor detection and removal. *IO*Shift can be seamlessly integrated into existing FL frameworks without adding any operation such as model modifications, enabling effortless deployment for backdoor defense.

- Extensive experiments show that across various Non-IID distribution and adversarial backdoor settings, *IO*Shift consistently outperforms SOTA backdoor defense methods.

## 2 RELATED WORK

**Backdoor Attack.** Backdoor attacks have become a well-studied security threat in federated learning. In these attacks, adversaries inject backdoors into the global model by modifying training samples and optimizing them using mini-batch SGD, a strategy known as the Vanilla attack Gu et al. (2019). To improve stealthiness and robustness, various advanced techniques have been proposed. PGD Wang et al. (2020) restricts malicious updates within a small perturbation range to avoid detection. Chameleon Dai & Li (2023) leverages contrastive learning to make backdoor samples visually blend in while remaining effective. DarkFed Li et al. (2024) introduces data-free backdoor attacks using shadow datasets, removing the dependency on local data. Recent attack methods leverage adversarial optimization Li et al. (2025a) to enhance the effectiveness and stealth of trigger design. A3FL Zhang et al. (2023) enhances backdoor persistence by incorporating adversarial training, mak-

ing the attack more resistant to model updates. In personalized FL, FPedBA Lyu et al. (2024) optimizes the loss function and aligning gradients to improve trigger injection. Mirages Li et al. (2025b) leverages both in-distribution and out-of-distribution data to adversarially optimize a trigger that induces an in-distribution mapping. Attackers employ distributed triggers to improve stealth and flexibility. DBA Xie et al. (2019) splits a global trigger into local ones, trained independently and jointly injected to evade detection. FCBA Liu et al. (2024b) enhances this by strategically combining local triggers. Other methods include Blend attacks Chen et al. (2017), which mix noise triggers with benign samples for subtle manipulation, and Semantic attacks Wang et al. (2020), which leverage natural features as triggers. Edge-case attacks Wang et al. (2020) focus on rare samples, boosting resistance to detection and preventing backdoor vanishing during training.

**Backdoor Defense.** To address backdoor attacks in federated learning, researchers have developed diverse defense mechanisms. Flame Nguyen et al. (2022) employs noise injection to neutralize backdoors, integrating HDBSCAN-based suspicious update detection and magnitude constraints to mitigate attacks. However, these methods face challenges in Non-IID environments, where benign updates may vary significantly, and sophisticated backdoor updates can mimic legitimate patterns. To address these limitations, Indicator Li & Dai (2024) introduces a "backdoor indicator task" during training, which rapidly decays in benign clients but remains stable in malicious clients injecting backdoors. FDCR Huang et al. (2024)detects potential attackers by estimating the Fisher information matrix, and adjusts the contribution of each client's update through reweighting. The recent method, AlignIns Xu et al. (2025), detects and mitigates malicious backdoor updates in federated learning by analyzing the directional characteristics of model updates on two levels: their alignment with the global model and the sign consistency of critical parameters. Other state-of-the-art methods are detailed in Appendix C.

## 3 THREAT MODEL

We consider a standard FL backdoor defense scenario, where the defender is the FL server.

**Attacker's Capabilities and Goals.** The attacker can compromise a subset of clients in federated learning, gaining full control over their local training, datasets, and model updates. Additionally, the attacker has access to the global model and historical updates, enabling them to adapt their strategies to enhance stealth while minimizing any negative impact on model accuracy. Notably, we do not restrict the attacker's choice of backdoor types for injection. The attacker's objective is to poison client updates so that the global model misclassifies inputs containing the trigger as the target class while evading anomaly detection defenses.

**Defender's Capabilities and Goals.** The defender aims to detect backdoors embedded in uploaded updates through a defense protocol. Although the defender lacks access to both the raw data and the data distribution of local clients, they do have white-box access to the model updates submitted by participating clients. Additionally, we assume the defender has no access to any data that shares the same distribution as the local clients' raw data. The defender's objective is to identify and remove malicious backdoor while preserving the global model's primary functionality.

## 4 METHODS

### 4.1 INTUITION OF *IO*SHIFT

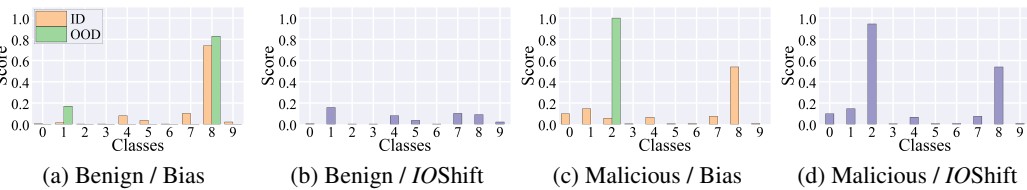

(a) Benign / Bias     (b) Benign / *IO*Shift     (c) Malicious / Bias     (d) Malicious / *IO*Shift

Figure 2: Model bias scores on ID and OOD data, and their *IO*Shift score under Dirichlet parameter $d = 0.1$ for benign and BadNets-implanted malicious models.

Before introducing *IO*Shift in detail, we first present the key insight that motivated its design. We focus on investigating how backdoor implantation affects the model inherent bias toward ID and OOD data under a centralized setting from the white-box perspective. Specifically, the evaluation is conducted using the widely used CIFAR10 dataset Krizhevsky et al. (2009) for classification task and the ResNet18 He et al. (2016) model architecture. The data distribution is simulated using a Dirichlet distribution with parameter $d$. The ID dataset consists of 200 images, selected by randomly choosing 20 images from each class in the CIFAR10 test set. For the OOD dataset, 200 images are randomly sampled from commonly used OOD dataset, 300K Random dataset Hendrycks et al. (2019).

Figure 2 illustrates the model bias and *IO*Shift scores for benign and BadNets-implanted malicious models with target class 2. The model bias score is computed as the mean soft label values across all samples in ID and OOD dataset, with the rationale and details explained in Section 4.2. We observe that when client data is skewed (Non-IID), the model exhibits a noticeable bias toward both ID and OOD data. Further, the implantation of a strong backdoor (Figure 2a to Figure 2c) forces the model to classify OOD data into the target class with extremely high confidence, while simultaneously reducing the bias toward ID data. The *IO*Shift score quantifies the shift in model bias between ID and OOD data after backdoor implantation. Notably, the bias shift for the target class is significantly larger than for other classes. These results in Figure 2b to Figure 2d reveal that, under appropriate evaluation, the strong associative activation paths introduced by backdoors effectively redirect the model bias toward the target class. This insight motivates our detection approach, which identifies malicious updates by analyzing the bias shift between ID and OOD data. All updates exhibiting a high bias discrepancy are flagged as malicious and further mitigated through adaptive backdoor removal. Figure 7 and Figure 8 in Appendix visualize scores under $d = 0.5$ and $d = 1$ (IID).

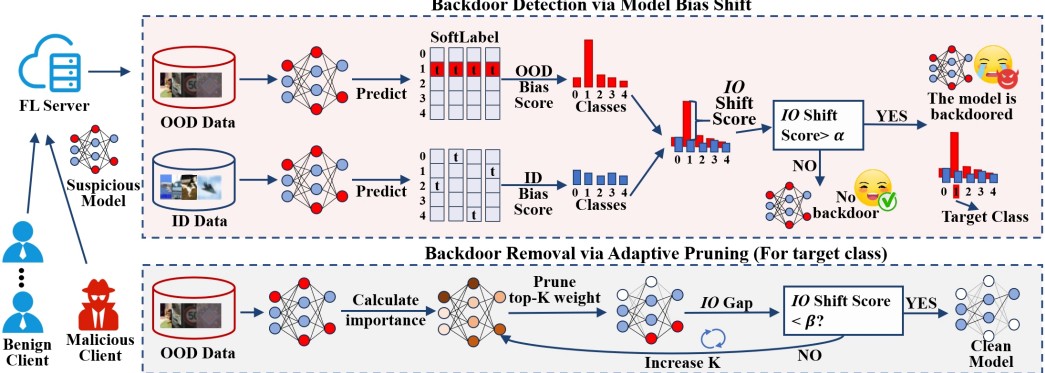

Figure 3: Workflow of *IO*Shift. (1) Backdoor detection via model bias shift; (2) Backdoor removal via adaptive pruning.

## 4.2 DETAILS OF *IO*SHIFT

Figure 3 and Algorithm 1 in Appendix illustrate the workflow of *IO*Shift. *IO*Shift consists of two main phases, aligned with defender's goals: 1) backdoor detection and 2) backdoor removal. Beyond the standard FL pipeline of model distribution, local training, update uploading, and global model aggregation, *IO*Shift introduces three key components: ID and OOD dataset preparation, *IO*Shift score computation, and adaptive backdoor removal. We detail each component in the following.

**ID and OOD dataset preparation.** To assess model bias, *IO*Shift requires both ID and OOD datasets. We assume the defender has access to a small amount of test data, such as 20 samples per class. This assumption is reasonable, as servers typically require test data to evaluate model generalization. Thus, ID dataset is represented by $D_I = x_i, y_i | i \in [1, M_I], M_I = K_I * N$, where $M_I$ is the total number, $K_I$ is the number of samples for each class and $N$ is the number of classes. OOD data lies outside the generalization scope of ID data and follows a different distribution. For example, in a CIFAR10 classification task, we construct the OOD dataset by randomly sampling from Tiny-ImageNet. Alternatively, unlabeled OOD samples can be drawn from public datasets or generated randomly using uniform distribution with a fixed seed, represented by $D_O = x_i | i \in [1, M_O]$.

***IO*Shift score computation.** Upon receiving update from client $j$, we convert the update into model parameters $\theta_j$ to evaluate its bias. Next, for the ID and OOD datasets, we compute three scores: ID bias score $BS^I$, OOD bias score $BS^O$, and *IO*Shift score $IO^S$. In simple terms, the ID bias score is computed as:

$$BS^I = \text{Ave}(SL(\theta_j, D_I)) = \frac{1}{M} \sum_{x_i \in D_I} SL(\theta_j, x_i),, \tag{1}$$

where $SL(\theta, \cdot)$ represents the soft labels produced by the model $\theta$ for sample $x$, and $\text{Ave}(\cdot)$ denotes the average function. The soft labels capture the probability distribution of the model bias toward the given data, reflecting its classification confidence de Vries & Thierens (2024); Wu et al. (2024). If a particular class consistently receives high confidence scores across all samples, the model is inherently biased toward this class. Similarly, we compute the OOD bias score for the OOD dataset:

$$BS^O = \text{Ave}(SL(\theta_j, D_O)) = \frac{1}{M} \sum_{x_I \in D_O} SL(\theta_j, x_i), \tag{2}$$

Then, the *IO*Shift score is given by:

$$IO^S = |BS^O - BS^I|. \tag{3}$$

At this stage, $IO^S$ is an $N$-dimensional vector, where each dimension represents the shift score for a specific class. If the shift score of any class exceeds a predefined threshold $\alpha$, we determine that the class has been backdoor-biased and mark it as the target class.

**Adaptive backdoor removal.** After detecting the target class $j$ and its corresponding shift score $IO_j^S$, our goal is to remove the backdoor by suppressing backdoor-related updates. However, the challenge lies in determining the minimal number of pruned neurons, $K$, that effectively eliminates the backdoor while preserving neurons essential for the clean task.

Guided by $IO_j^S$, we assume that once $IO_j^S$ falls below a removal threshold $\beta$, the backdoor effect has been mitigated to an acceptable level. The specific relationship between backdoor ASR and the threshold $\beta$ is detailed in Figure 6. $\beta$ is obtained by average IOShfit score from the first 30 epochs of the number of epochs in which the backdoor occurs. To determine the order of neurons to prune, we estimate neuron importance using the Fisher Information Matrix on the OOD dataset $D_O$ and rank neurons from most to least important:

$$I_w^O = (\frac{\partial \mathcal{L}(D_O, \theta_j)}{\partial w})^2. \tag{4}$$

This approach is based on the observation that if a backdoor path exists, OOD data will also activate this shortcut path. Thus, we prioritize pruning neurons that are highly important for the backdoor task. In summary, we initialize $K$ and iteratively increase it, pruning the top-$K$ gradient updates until the predefined stopping condition is met. Note that the pruning operation targets malicious updates. If applied during model retrospection, direct neuron pruning can be performed instead. Empirical results demonstrate that both approaches yield similar performance.

## 5 EXPERIMENTS

### 5.1 EXPERIMENTAL SETTINGS

**Datasets and Models.** We conduct comprehensive evaluations of *IO*Shift on two benchmark datasets: CIFAR10 Krizhevsky et al. (2009) and Tiny-ImageNet Le & Yang (2015). Our experiments utilize three representative neural architectures: ResNet18 He et al. (2016), VGG16 Simonyan (2014) and ResNet50 He et al. (2016) (Appendix C).

**FL setup.** By default, we set the total number of clients to $N = 100$, with the server randomly selecting $M = 10$ (10%) clients per each epoch for model updates. To simulate realistic federated learning scenarios, we distribute each dataset across clients in a Non-IID manner using a Dirichlet distribution-basedHsu et al. (2019) sampling strategy. The concentration parameter $d$ is set to 0.1, creating a challenging testing environment with significant differences in client data distribution.

Additionally, to thoroughly evaluate the robustness of *IO*Shift, we conduct further experiments under various Non-IID conditions by adjusting the Dirichlet concentration parameter, simulating different levels of data heterogeneity among clients.

**Attack Settings.** We evaluate *IO*Shift against seven types of backdoor attacks, including three fixed-trigger attacks: Vanilla Gu et al. (2019), PGD Wang et al. (2020), ChameleonDai & Li (2023), data-free attack: DarkFed Li et al. (2024) and there adversarial trigger attacks: A3FL Zhang et al. (2023), PFedBA Lyu et al. (2024), Mirages Li et al. (2025b). To ensure optimal attack performance, we configure these attacks following their original settings and released code as closely as possible. By default, we consider a FL setting with $B = 10$ malicious clients among $N = 100$ total clients. At each epoch, FL server randomly select ten clients to participant updates. The malicious clients can only perform attacks during the FL epochs in which they are selected. For CIFAR10, we set the trigger size to 3×3, and for Tiny-ImageNet, we set the trigger size to 6×6.

**Defense Settings.** We compare *IO*Shift with eight state-of-the-art defenses: Flame Nguyen et al. (2022), FDCR Huang et al. (2024), Indicator Li & Dai (2024), AlignIns Xu et al. (2025), as well as Multi-Krum Blanchard et al. (2017), Deepsight Rieger et al. (2022), Foolsgold Fung et al. (2018), Rflabt Wang et al. (2022) (Appendix C). Also, we compare *IO*Shift with one model recovery approaches: FedRecover Cao et al. (2023). For a fair comparison, we closely follow their original settings based on their released codes. For *IO*Shift, we set the $\alpha$ to 0.8 and $\beta$ to 0.5, with $K$ increasing by 0.5% per step. We randomly selected 1,000 images from the Random Image dataset to serve as the OOD dataset. The ID dataset consists of 20 images per class, which is sufficiently small to meet practical requirements. For models with batch normalization (BN) layers, we estimate the mean and standard deviation using the OOD dataset and the global model to mitigate statistical biases during inference caused by BN layers.

**Evaluation Metrics.** We use accuracy (ACC) to assess model performance on clean data and attack success rate (ASR) to evaluate the backdoor effect, with higher ASR indicating a stronger attack. To evaluate defense performance, we employ true positive rate (TPR) and false positive rate (FPR) Qi et al. (2023b). TPR is calculated as the ratio of correctly identified malicious updates to the total number of malicious updates; FPR is computed as the ratio of benign clients mistakenly classified as malicious to the total number of benign updates.

## 5.2 EXPERIMENT RESULTS

**Detection Performance.** Table 1 shows the comparison of *IO*Shift with four SOTA backdoor defenses on TPR, FPR and ASR under different attacks and training epochs. Visualizations of the corresponding *IO*Shift scores are shown in Figure 9 in Appendix. The Dirichlet parameter $d$ is set to 0.1. The attacker starts attacking at global epochs 400, 800 or 1200, representing different training stages of clean task. The attack lasts for 100 global epochs, and we report overall performance metrics. Compared to other methods, *IO*Shift achieves the highest TPR, the lowest FPR and ASR across all attack methods, datasets, and attack epochs. This is because the strong activation path of backdoors significantly shift model bias. For DarkFed, A3FL, PFedBA and Mirages the TPR remains close to 100%, highlighting the effectiveness of recent adversarial backdoor attacks to shift model bias. However, ASR does not reach zero because *IO*Shift relies on *IO*Shift scores, which cannot directly capture the exact trigger pattern. Notably, Mirages is particularly difficult to eliminate, as it leverages both in-distribution and out-of-distribution data to optimize its trigger, thereby constructing a backdoor path that remains within the in-distribution mapping space, rendering it highly stealthy and resistant to detection. Nevertheless, when ASR falls below 15, we consider the attack to be effectively failed. The results indicate that under highly non-IID scenarios, Flame is largely ineffective against existing attacks—even the most basic ones such as the vanilla attack. Furthermore, Flame's effectiveness is worse than no protection at all. This is because its FPR is relatively high, resulting in the removal of numerous benign clients and consequently increasing the proportion of malicious clients. While FDCR and AlignIns show relatively good performance in defending against simple attacks, they struggle to detect more recent and sophisticated trigger-optimization-based attacks. This limitation arises because these defenses rely heavily on statistical indicators such as norm-based metrics and cosine similarity. However, under highly non-IID scenarios, even benign updates can exhibit significant divergence, making it difficult to reliably distinguish between malicious and benign clients using these metrics.

Table 1: Comparison of TPR, FPR and ASR under different defenses, $d = 0.1$.

| Datasets | Attack | Epoch | No Defense | Flame | | | FDCR | | | Indicator | | | AlignIns | | | IOShift | | |
|---|---|---|---|---|---|---|---|---|---|---|---|---|---|---|---|---|---|---|
| | | | ASR | TPR | FPR | ASR | TPR | FPR | ASR | TPR | FPR | ASR | TPR | FPR | ASR | TPR | FPR | ASR |
| CIFAR10 | Vanilla | 400 | 46.7 | 17.0 | 38.1 | 61.9 | 87.0 | 26.8 | 19.2 | 88.0 | 48.9 | 18.6 | 90.0 | 45.9 | 12.5 | 95.0 | 2.4 | 9.8 |
| | | 800 | 52.4 | 21.0 | 34.6 | 70.2 | 88.0 | 29.6 | 12.1 | 90.0 | 44.0 | 13.4 | 90.0 | 44.8 | 12.8 | 95.0 | 3.2 | 10.2 |
| | | 1200 | 70.5 | 18.0 | 36.2 | 76.5 | 90.0 | 28.4 | 11.2 | 93.0 | 46.3 | 11.2 | 92.0 | 44.2 | 11.8 | 95.0 | 2.8 | 9.2 |
| | PGD | 400 | 48.4 | 10.0 | 40.8 | 55.9 | 75.0 | 29.1 | 26.9 | 80.0 | 42.8 | 21.4 | 48.0 | 23.4 | 62.8 | 93.0 | 4.2 | 11.2 |
| | | 800 | 51.1 | 12.0 | 32.8 | 70.9 | 78.0 | 28.1 | 24.3 | 88.0 | 46.7 | 12.8 | 49.0 | 24.5 | 63.9 | 94.0 | 2.2 | 10.9 |
| | | 1200 | 67.3 | 15.0 | 32.0 | 73.8 | 80.0 | 29.3 | 22.1 | 90.0 | 41.3 | 10.8 | 54.0 | 25.8 | 61.0 | 96.0 | 3.1 | 9.9 |
| | Chameleon | 400 | 50.2 | 14.0 | 31.4 | 56.2 | 70.0 | 28.4 | 34.8 | 90.0 | 41.4 | 20.2 | 82.0 | 22.8 | 32.9 | 94.0 | 2.8 | 10.2 |
| | | 800 | 62.6 | 19.0 | 29.0 | 74.8 | 72.0 | 28.9 | 33.1 | 92.0 | 42.0 | 11.1 | 84.0 | 21.2 | 30.8 | 94.0 | 2.3 | 10.2 |
| | | 1200 | 71.9 | 14.0 | 36.4 | 75.8 | 76.0 | 27.5 | 30.2 | 92.0 | 43.4 | 11.9 | 88.0 | 20.1 | 24.5 | 95.0 | 2.1 | 9.6 |
| | DarkFed | 400 | 65.5 | 13.0 | 33.8 | 70.1 | 82.0 | 25.1 | 22.0 | 92.0 | 38.9 | 13.8 | 74.0 | 24.8 | 41.8 | 98.0 | 2.4 | 9.8 |
| | | 800 | 80.1 | 11.0 | 28.9 | 84.2 | 85.0 | 27.8 | 20.1 | 94.0 | 43.4 | 13.0 | 78.0 | 25.1 | 38.2 | 100.0 | 3.4 | 9.1 |
| | | 1200 | 88.2 | 13.0 | 32.9 | 89.1 | 88.0 | 29.1 | 17.9 | 94.0 | 46.0 | 15.2 | 82.0 | 25.4 | 34.8 | 100.0 | 4.5 | 9.2 |
| | A3FL | 400 | 94.5 | 8.0 | 38.4 | 96.5 | 71.0 | 28.9 | 56.8 | 92.0 | 39.0 | 34.5 | 38.0 | 25.8 | 87.9 | 100.0 | 2.0 | 14.0 |
| | | 800 | 96.9 | 8.0 | 38.2 | 97.8 | 72.0 | 29.8 | 58.9 | 93.0 | 39.0 | 37.8 | 40.0 | 25.1 | 84.2 | 100.0 | 2.6 | 14.5 |
| | | 1200 | 100 | 6.0 | 37.8 | 98.0 | 76.0 | 28.9 | 52.1 | 92.0 | 41.0 | 38.0 | 46.0 | 25.9 | 81.1 | 100.0 | 1.1 | 14.6 |
| | PFedBA | 400 | 90.1 | 4.0 | 39.6 | 91.5 | 73.0 | 28.8 | 55.2 | 90.0 | 42.8 | 26.1 | 42.0 | 24.8 | 76.5 | 100.0 | 2.1 | 13.7 |
| | | 800 | 93.5 | 3.0 | 39.2 | 95.2 | 76.0 | 29.1 | 51.1 | 91.0 | 45.1 | 26.4 | 45.0 | 26.0 | 72.1 | 100.0 | 1.9 | 14.8 |
| | | 1200 | 100 | 3.0 | 39.9 | 99.2 | 77.0 | 27.4 | 49.8 | 92.0 | 43.5 | 30.5 | 48.0 | 25.1 | 70.5 | 100.0 | 2.6 | 13.9 |
| | Mirages | 400 | 100.0 | 17.0 | 38.1 | 61.9 | 40.0 | 27.9 | 88.5 | 88.0 | 48.9 | 18.6 | 37.0 | 26.2 | 90.5 | 100.0 | 1.8 | 5.0 |
| | | 800 | 100.0 | 17.0 | 38.1 | 61.9 | 36.0 | 27.8 | 90.2 | 88.0 | 48.9 | 18.6 | 41.0 | 25.1 | 88.1 | 100.0 | 1.8 | 4.9 |
| | | 1200 | 100.0 | 17.0 | 38.1 | 61.9 | 41.0 | 28.6 | 89.6 | 88.0 | 48.9 | 18.6 | 43.0 | 26.0 | 84.1 | 100.0 | 2.0 | 4.2 |
| Tiny-ImageNet | Vanilla | 800 | 57.5 | 0.0 | 40.4 | 66.5 | 66.0 | 33.6 | 32.5 | 74.0 | 38.2 | 27.4 | 78.0 | 36.0 | 20.2 | 93.0 | 1.6 | 9.5 |
| | | 1200 | 70.2 | 0.0 | 42.5 | 68.3 | 69.0 | 34.8 | 33.8 | 79.0 | 41.5 | 29.4 | 82.0 | 35.4 | 18.1 | 95.0 | 2.0 | 5.0 |
| | | 1600 | 81.4 | 0.0 | 41.6 | 75.9 | 68.0 | 36.2 | 34.1 | 84.0 | 42.1 | 38.2 | 84.0 | 36.5 | 16.1 | 95.0 | 1.5 | 4.7 |
| | PGD | 800 | 54.1 | 0.0 | 42.1 | 60.1 | 54.0 | 39.2 | 42.0 | 69.0 | 41.2 | 22.9 | 44.0 | 37.2 | 44.8 | 91.0 | 1.9 | 8.4 |
| | | 1200 | 67.8 | 0.0 | 44.9 | 67.1 | 51.0 | 38.4 | 46.1 | 76.0 | 43.1 | 15.9 | 46.0 | 36.8 | 42.1 | 92.0 | 2.1 | 7.1 |
| | | 1600 | 75.9 | 0.0 | 44.6 | 71.1 | 50.0 | 39.9 | 48.5 | 82.0 | 39.8 | 12.5 | 47.0 | 36.2 | 41.5 | 92.0 | 2.0 | 5.2 |
| | Chameleon | 800 | 58.45 | 3.0 | 42.8 | 64.8 | 53.0 | 38.1 | 44.1 | 70.0 | 40.5 | 34.1 | 70.0 | 35.2 | 28.2 | 96.0 | 2.3 | 4.3 |
| | | 1200 | 72.6 | 2.0 | 43.0 | 70.8 | 55.0 | 39.6 | 45.8 | 74.0 | 42.5 | 36.2 | 74.0 | 35.6 | 25.1 | 98.0 | 2.2 | 1.2 |
| | | 1600 | 84.1 | 3.0 | 43.2 | 86.9 | 55.0 | 38.1 | 48.3 | 75.0 | 42.2 | 39.2 | 75.0 | 35.8 | 22.5 | 100.0 | 2.1 | 0.6 |
| | DarkFed | 800 | 74.2 | 6.0 | 41.5 | 76.1 | 54.0 | 37.8 | 42.5 | 72.0 | 41.6 | 38.2 | 77.0 | 35.2 | 21.5 | 100.0 | 1.9 | 0.4 |
| | | 1200 | 85.6 | 7.0 | 40.5 | 84.3 | 56.0 | 38.2 | 41.8 | 76.0 | 41.1 | 41.5 | 79.0 | 34.8 | 20.8 | 100.0 | 2.2 | 0.6 |
| | | 1600 | 90.1 | 7.0 | 42.6 | 90.4 | 55.0 | 38.1 | 43.1 | 78.0 | 42.0 | 41.2 | 82.0 | 33.1 | 19.5 | 100.0 | 2.0 | 0.5 |
| | A3FL | 800 | 99.5 | 0.0 | 43.2 | 99.3 | 44.0 | 37.7 | 74.8 | 73.0 | 41.2 | 78.5 | 40.0 | 37.5 | 76.2 | 100.0 | 1.7 | 6.5 |
| | | 1200 | 98.8 | 0.0 | 41.5 | 98.5 | 48.0 | 38.1 | 76.5 | 75.0 | 42.0 | 72.6 | 42.0 | 37.1 | 75.2 | 100.0 | 2.1 | 6.3 |
| | | 1600 | 100.0 | 0.0 | 41.8 | 99.6 | 50.0 | 38.9 | 80.5 | 75.0 | 39.8 | 73.8 | 46.0 | 36.5 | 70.1 | 100.0 | 1.9 | 5.7 |
| | PFedBA | 800 | 96.2 | 0.0 | 41.6 | 99.4 | 48.0 | 35.2 | 68.9 | 72.0 | 42.1 | 71.5 | 41.0 | 37.1 | 75.2 | 100.0 | 2.1 | 6.6 |
| | | 1200 | 97.6 | 0.0 | 40.8 | 99.3 | 49.0 | 35.1 | 73.5 | 78.0 | 41.9 | 64.3 | 43.0 | 36.9 | 73.1 | 100.0 | 2.3 | 5.8 |
| | | 1600 | 99.5 | 0.0 | 40.4 | 98.8 | 52.0 | 35.8 | 83.5 | 76.0 | 41.2 | 68.9 | 46.0 | 35.1 | 70.5 | 100.0 | 2.0 | 5.4 |
| | Mirages | 400 | 100.0 | 17.0 | 38.1 | 61.9 | 42.0 | 34.8 | 88.2 | 88.0 | 48.9 | 18.6 | 36.0 | 36.2 | 82.0 | 100.0 | 2.0 | 5.1 |
| | | 800 | 100.0 | 17.0 | 38.1 | 61.9 | 45.0 | 35.6 | 90.5 | 88.0 | 48.9 | 18.6 | 38.0 | 35.9 | 80.1 | 100.0 | 1.9 | 5.5 |
| | | 1200 | 100.0 | 17.0 | 38.1 | 61.9 | 44.0 | 35.1 | 90.2 | 88.0 | 48.9 | 18.6 | 42.0 | 36.1 | 79.2 | 100.0 | 2.1 | 5.9 |

In contrast, Indicator injects an active backdoor to facilitate defense and generally achieves a higher TPR compared to other methods. Nonetheless, its performance still falls short of *IO*Shift. Moreover, Indicator suffers from a notably high FPR. This is because, although the active backdoor improves detection capabilities, it tends to misclassify benign clients—particularly those with highly skewed local data distributions—as malicious, thereby significantly increasing the FPR.

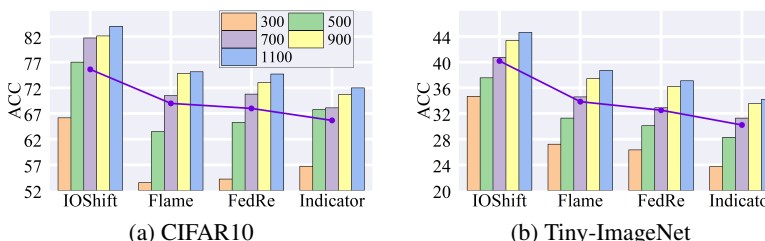

(a) CIFAR10    (b) Tiny-ImageNet

Figure 4: Comparison of removal performance on Accuracy (ACC) at different global training epochs and different defenses under Vanilla attack.

**Removal Performance.** Figure 4 shows the comparison of accuracy of global model and SOTA methods at different training epochs under Vanilla attack. The performances under A3FL attack is shown in Figure 12 in Appendix. The purple dashed line represents the average value across training epochs. For FedRecover, we apply backdoor removal based on the current best-performing detection method, Indicator. The results show that *IO*Shift consistently achieves higher accuracy than other recovery methods across all training epochs, with an average improvement of 9% on CIFAR10 task. This advantage stems from our adaptive pruning strategy, which automatically determines the minimal pruning required to eliminate ASR while minimizing the impact on the clean task. The lowest

performance observed with Indicator highlights the necessity of backdoor removal, demonstrating that targeted pruning is more effective than simply discarding updates.

## 5.3 IMPACT OF HYPERPARAMETERS

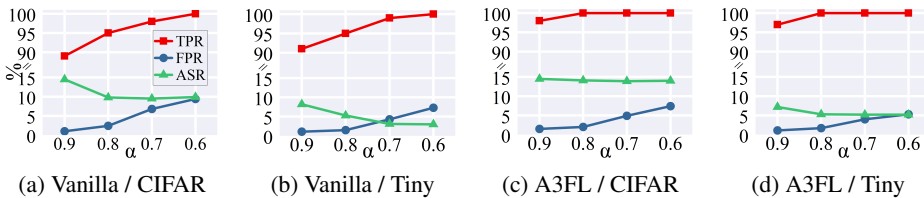

|  (a) Vanilla / CIFAR | (b) Vanilla / Tiny | (c) A3FL / CIFAR | (d) A3FL / Tiny |

Figure 5: Performance of *IO*Shift under different detection threshold $\alpha$.

**Different Detection Threshold $\alpha$.** Figure 5 shows the performance of *IO*Shift under different detection threshold $\alpha$. Overall, reducing $\alpha$ slightly increases the TPR of *IO*Shift. For the A3FL attack, when $\alpha = 0.8$, TPR reaches nearly 99%. In contrast, for CIFAR10, TPR is limited to 90% because the adversarial method in A3FL causes the model bias on OOD data to exceed 0.8, which also explains its higher ASR. In general, ASR follows the same trend as TPR: when detection rates are high, ASR can be fully eliminated. However, as $\alpha$ increases, FPR rises significantly. This is because some benign clients with skewed data distributions may exhibit random ID bias scores, though this does not affect the effectiveness of ASR suppression.

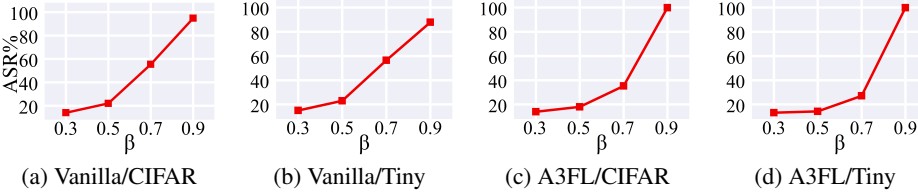

| (a) Vanilla/CIFAR | (b) Vanilla/Tiny | (c) A3FL/CIFAR | (d) A3FL/Tiny |

Figure 6: Relationship between ASR and $\beta$.

**Relationship between ASR and $\beta$.** Figure 6 illustrates the relationship between ASR and the threshold $\beta$ during backdoor removal via adaptive pruning. The results show that as $\beta$ increases, ASR also rises, aligning with the principles underlying our backdoor detection approach. This is because the effectiveness of a backdoor stems from embedding a strong activation path, which significantly biases the model. Notably, when $\beta$ is below 0.5, the model's ASR drops to 40%, which we consider the threshold for an unsuccessful backdoor attack.

## 5.4 FURTHER UNDERSTANDING

Table 2: Performance of *IO*Shift under different sources of OOD data.

| OOD Source | Vanilla | | | | | | A3FL | | | | | |
| | CIFAR10 | | | Tiny-ImageNet | | | CIFAR10 | | | Tiny-ImageNet | | |
| | TPR | FPR | ASR | TPR | FPR | ASR | TPR | FPR | ASR | TPR | FPR | ASR |
| **Random** | 95.0 | 2.4 | 9.8 | 95.0 | 1.5 | 0.7 | 100.0 | 2.0 | 16.0 | 100.0 | 1.7 | 6.2 |
| **CIFAR100** | 93.0 | 3.4 | 9.9 | 92.0 | 2.6 | 1.1 | 100.0 | 2.6 | 15.7 | 100.0 | 2.0 | 6.4 |
| **Noise** | 93.0 | 5.9 | 10.5 | 92.0 | 7.9 | 1.2 | 98.0 | 3.2 | 18.2 | 100.0 | 2.8 | 6.7 |
| **GTSRB** | 92.0 | 5.5 | 10.4 | 91.0 | 6.3 | 1.2 | 98.0 | 2.7 | 17.8 | 99.0 | 2.2 | 7.5 |

**Different Sources of OOD data.** Table 2 shows the performance of *IO*Shift with different sources of OOD data under CIFAR10 and Tiny-ImageNet datasets. The 300K Random dataset Hendrycks et al. (2019) is a widely used dataset for outlier exposure, which is a subset from 80 Million Tiny Images. For Noise, we generate samples using a fixed seed following a uniform distribution. For GTSRB, images are converted into corresponding pixel representations with clean tasks. The results

indicate that detection performance is not sensitive to the choice of OOD data source, whether on CIFAR10 or Tiny-ImageNet tasks. This is because all these datasets are out-of-distribution, making it difficult for the model to make reliable predictions, as discussed in He & Garcia (2009).

Table 3: Performance on TPR, FPR and ASR under different numbers of compromised clients.

| Datasets | Num% | Vanilla | | | PGD | | | Chameleon | | | DarkFed | | | A3FL | | | PFedBA | | | Mirages | | |
|---|---|---|---|---|---|---|---|---|---|---|---|---|---|---|---|---|---|---|---|---|---|---|
| | | TPR | FPR | ASR | TPR | FPR | ASR | TPR | FPR | ASR | TPR | FPR | ASR | TPR | FPR | ASR | TPR | FPR | ASR | TPR | FPR | ASR |
| CIFAR10 | 5% | 96.0 | 3.2 | 8.6 | 95.0 | 2.9 | 8.7 | 96.0 | 2.6 | 9.1 | 100.0 | 3.6 | 9.2 | 100.0 | 1.6 | 13.7 | 100.0 | 2.0 | 13.5 | 100.0 | 2.1 | 12.1 |
| | 10% | 95.0 | 2.8 | 9.2 | 96.0 | 3.1 | 9.9 | 95.0 | 2.1 | 9.6 | 100.0 | 4.5 | 9.2 | 100 | 1.1 | 14.6 | 100.0 | 2.6 | 13.9 | 100.0 | 1.8 | 13.9 |
| | 20% | 95.0 | 1.5 | 12.1 | 94.5 | 2.2 | 12.1 | 95.5 | 1.8 | 10.5 | 100.0 | 3.6 | 8.9 | 97.5 | 1.0 | 16.2 | 97.0 | 2.2 | 14.1 | 98.0 | 2.0 | 12.6 |
| Tiny-ImageNet | 5% | 94.0 | 2.0 | 5.2 | 94.0 | 2.3 | 4.8 | 98.0 | 1.9 | 2.6 | 100.0 | 2.3 | 0.9 | 100.0 | 2.2 | 5.9 | 100.0 | 2.1 | 5.9 | 100.0 | 1.6 | 5.8 |
| | 10% | 95.0 | 1.5 | 4.7 | 92.0 | 2.0 | 5.2 | 100.0 | 2.1 | 0.6 | 100.0 | 2.0 | 0.5 | 100.0 | 1.9 | 5.7 | 100.0 | 2.0 | 5.4 | 100 | 1.9 | 5.5 |
| | 20% | 94.0 | 1.0 | 8.6 | 90.0 | 2.0 | 8.5 | 99.0 | 1.6 | 2.1 | 100.0 | 1.6 | 1.2 | 99.0 | 1.2 | 7.8 | 98.5 | 1.6 | 6.2 | 99.0 | 1.8 | 5.1 |

**Different Numbers of Compromised Clients.** Table 3 shows the performance of *IO*Shift in a federated learning system with varying proportions of attackers. Results indicate that *IO*Shift is robust to varying numbers of attackers. This robustness stems from *IO*Shift's individualized detection approach, which leverages the Backdoor-Induced Model Bias Shift phenomenon on a per-model basis. Compared to existing defenses based on anomaly detection or benign parameter space estimation, *IO*Shift offers greater resilience.

**Advanced Backdoor Attack Types.** Table 4 shows the performance of *IO*Shift against advanced backdoor attack types. The results demonstrate that *IO*Shift maintains significant detection and removal effectiveness even against more advanced distributed backdoor attacks. This is because, whether the attack is distributed or centralized, as long as the local ASR is high, it inevitably biases the model. For Semantic and Blend attacks, *IO*Shift achieves TPR of 74% and 87%, respectively. The reason is that trigger in Semantic attack has subtle features within the benign data space, causing partial overlap between the backdoor and benign activation paths. Similarity, Blend attack introduce small perturbations across the entire input space, leading to a more concealed model bias shift.

Table 4: Performance of *IO*Shift against advanced backdoor types.

| Dataset | Type | TPR | FPR | ASR |
|---|---|---|---|---|
| CIFAR10 | FCBA | 99.0 | 3.7 | 8.2 |
| | DBA | 99.0 | 4.1 | 7.9 |
| | Semantic | 74.0 | 3.6 | 21.5 |
| | Blend | 87.0 | 2.2 | 14.1 |
| | Edge | 79.0 | 4.1 | 8.1 |
| | WaNet | 59.0 | 2.2 | 14.2 |
| | A_Blend | 73.0 | 3.6 | 21.8 |
| Tiny-ImageNet | FCBA | 98.0 | 2.5 | 5.1 |
| | DBA | 95.0 | 2.9 | 6.9 |
| | Blend | 84.0 | 2.8 | 11.2 |
| | WaNet | 68.0 | 2.4 | 1.2 |
| | A_Blend | 65.0 | 3.2 | 2.2 |

Note that in our experimental setting, WaNet Nguyen & Tran (2021) and A_Blend Qi et al. (2023a), which perform well in centralized training scenarios, can hardly be considered as a successful attack in distributed scenarios.

**Other Experiments.** Experiments on different Dirichlet settings, poisoned learning rates, different network architectures, backdoor removal performance, computational costs, visualization of *IO*Shift scores are detailed in Appendix C.

# 6 CONCLUSION

In this paper, we introduce the *Backdoor-Induced Model Bias Shift* phenomenon, where backdoors introduce a stronger malicious activation path between the trigger and the target class, leading to an absolute bias toward OOD data. Inspired by this observation, we propose *IO*Shift, a federated backdoor detection and removal framework based on model bias shift between ID and OOD data. Specifically, if an uploaded model exhibits a significant bias shift toward a particular class, it indicates that a strong backdoor activation has been embedded in that class. *IO*Shift breaks away from the strong assumption of existing defenses that rely on defining a benign parameter space, making it more effective in FL with high Non-IID degree. Moreover, *IO*Shift can be seamlessly deployed in existing FL frameworks by adding detection and removal steps without modifying the core framework. Extensive experiments demonstrate that *IO*Shift outperforms existing methods in both backdoor detection and removal, achieving superior performance across various settings.

**Limitations.** One limitation of *IO*Shift is that we still require a very small set of clean ID samples (e.g., 20 samples per class). Despite being small, it still introduces an extra capability of defender. In addition, based on the better performances, our running time is a bit more than the gradient checking scheme. However, these times are essentially negligible compared to the client training time.

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
