**IOShift: Backdoor Defense via Model Bias Shift in Federated Learning**
Supplementary Material

## A  BROADER IMPACTS

Federated learning, while offering strong data protection, is particularly vulnerable to sophisticated security threats such as backdoor attacks. These attacks can lead to unforeseen and potentially harmful consequences in real-world applications. This work provides a deep investigation into the underlying mechanisms that contribute to the success of backdoor attacks in federated learning systems and introduces *IO*Shift, a novel defense framework grounded in the observation of Backdoor-Induced Model Bias Shift phenomenon. *IO*Shift can be seamlessly integrated into existing federated learning infrastructures, enhancing their security without compromising performance. By strengthening the resilience of federated learning against malicious interference, this work contributes to the development of more robust and trustworthy distributed learning systems. Ultimately, it promotes safer adoption of AI technologies and facilitates the secure flow of knowledge across different sectors and institutions, yielding positive impacts on data-driven decision-making and collaborative innovation across society.

---

**Algorithm 1** Overview of *IO*Shift

---

**Input**: Clients $\{C_i\}$ and their data $\{D_i\}$, dataset size $\kappa$, detection threshold $\alpha$ and $\beta$.
**Output**: Global model at epoch $t$, $\theta_G^{(T)}$.
  1: Server prepares ID dataset $\mathcal{D}_I$ and OOD dataset $\mathcal{D}_O$ with size $\kappa$
  2: **for** epoch $e$ to $T$ **do**
  3:     Server selects $M$ clients and distributes $\theta_G^{(e)}$
         **/* Client Local Training */**
  4:     Client $i$ trains $\theta_G^{(e+1)}$ by $D_i$, uploads updates $g_i^{(e)}$
         **/* Backdoor Detection */**
  5:     Server computes ID bias score $BS^I$ for $\mathcal{D}_I$ and OOD bias score $BS^O$ for $\mathcal{D}_O$ by Equation 1 and Equation 2
  6:     Server gets *IO*Shift score $IO^S = \parallel BS^O - BS^I \parallel$
  7:     **if** $IO_j^S \in IO^S > \alpha$ **then**
  8:        Label class $j$ as target class
  9:     **end if**
         **/* Backdoor Removal */**
10:     **while** $IO_j^S < \beta$ **do**
11:        Server labels the class of samples in $\mathcal{D}_O$ as $j$
12:        Server computes neuron importance by Equation 4
13:        Server prunes top $K$ neurons in $g_i^{(e)}$ and recalculates $IO_j^S$
14:        $K + +$
15:     **end while**
16:     Server updates global model
17: **end for**
18: **return** Global model at epoch $t$: $\theta_G^{(T)}$.

---

## B  VISUALIZATION OF MODEL BIAS SCORES

Figure 7 and Figure 8 visualize the model bias and *IO*Shift score under $d = 0.5$ and $d = 1$ (IID). The results confirm that backdoors indeed shift *IO*Shift score, regardless of whether the data distribution is IID or Non-IID.

## C  OTHER EXPERIMENTS

**Other Detection Performance.**  Table 5 shows the performance results with Multi-Krum Blanchard et al. (2017), Deepsight Rieger et al. (2022), Foolsgold Fung et al. (2018), Rflabt Wang et al. (2022).

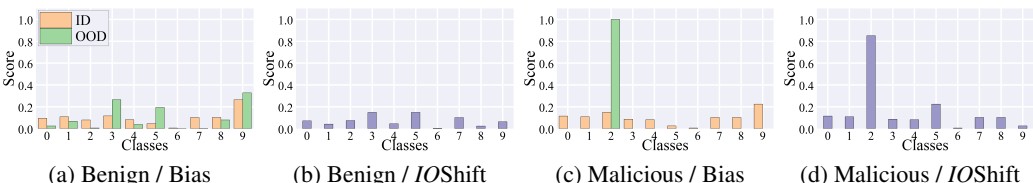

(a) Benign / Bias     (b) Benign / *IO*Shift     (c) Malicious / Bias     (d) Malicious / *IO*Shift

Figure 7: Model bias scores on ID and OOD data, and their *IO*Shift scores under Dirichlet parameter $d = 0.5$ for benign and BadNets-implanted malicious models.

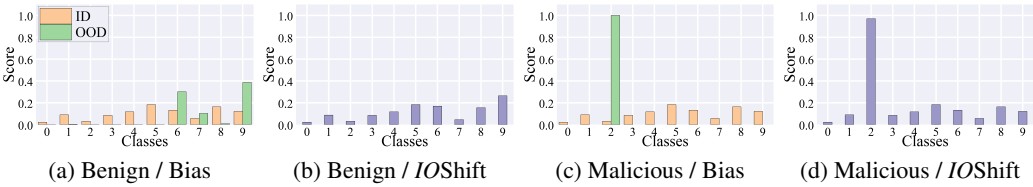

(a) Benign / Bias     (b) Benign / *IO*Shift     (c) Malicious / Bias     (d) Malicious / *IO*Shift

Figure 8: Model bias scores on ID and OOD data, and their *IO*Shift scores under Dirichlet parameter $d = 1$ (IID) for benign and BadNets-implanted malicious models.

Multi-Krum Blanchard et al. (2017) selects client updates with minimal Euclidean distance to filter out malicious contributions. Foolsgold Fung et al. (2018) differentiates attackers by assessing update diversity, as benign clients typically exhibit greater variability. Deepsight Rieger et al. (2022) identifies backdoor patterns by analyzing subtle model update behaviors, assuming that backdoored models reflect distinct training data characteristics. RFLBAT Wang et al. (2022) enhances detection using dimensionality reduction to amplify differences between benign and malicious updates, followed by a two-stage filtering process with outlier removal and cluster analysis.

Comparison of TPR, FPR and ASR under different defenses, $d = 0.1$. From the results, existing anomaly detection-based methods struggle to identify malicious updates, with TPRs falling below 20%. Specifically, Multi-Krum and RFLBAT fail to detect any malicious updates in most cases. This is because, under highly Non-IID settings, defining a reliable benign parameter space from client updates becomes infeasible. Consequently, these methods fail in both detection and removal, leading to persistently high ASRs. As for the FPR in benign update detection, most existing methods exhibit extremely high misclassification rates. In particular, Indicator and Multi-Krum exceed 40%, meaning that nearly half of the benign clients are excluded in each epoch, resulting in significant resource waste. The relatively lower FPRs of Deepsight and RFLBAT stem from their limited number of flagged anomalies. However, given their extremely low TPRs, the updates they classify as anomalies are predominantly benign. Although Indicator achieves a moderately high TPR, it suffers from an excessively high FPR. This occurs because, under low Dirichlet parameters, model biases on benign samples strengthen the defense-oriented Indicator Task embedded by the defender in advance.

**Different Dirichlet Parameters $d$ and Poisoned Learning Rate.** Table 6 shows the performance of *IO*Shift under different backdoor attacks with varying Dirichlet parameters $d$ and poisoned learning rate (plr). The attackers can manually adjust poisoned learning rate to control the magnitude of malicious updates, enabling them to evade existing anomaly detection methods. The learning rate of benign client is set to 0.05. Extensive experimental results show that regardless of plr adjustments, as long as the attack success rate reaches 60%, *IO*Shift maintains stable performance.

**Visualization of *IO*Shift Scores.** Figure 9 shows *IO*Shift scores corresponding to Table 1. The high *IO*Shift score in target class 2 indicates why *IO*Shift is able to succeed.

**Different Sizes of OOD Dataset.** Figure 10 shows the performance of *IO*Shift under different sizes of OOD dataset. The results show that as the size of OOD dataset increases, *IO*Shift achieves a higher TPR, lower FPR, and reduced ASR. This is because a larger OOD dataset allows for a more precise separation of backdoor-induced biases from the model's genuine features. As the sample size exceeds 1000, the ASR is almost entirely suppressed, indicating that *IO*Shift can effectively

Table 5: Comparison of TPR, FPR and ASR under different defenses, $d = 0.1$.

| Datasets | Attack | Epoch | No Defense | Multi-Krum | | | Deepsight | | | Foolsgold | | | RFLBAT | | |
|---|---|---|---|---|---|---|---|---|---|---|---|---|---|---|---|
| | | | ASR | TPR | FPR | ASR | TPR | FPR | ASR | TPR | FPR | ASR | TPR | FPR | ASR |
| CIFAR 10 | Vanilla | 400 | 46.7 | 0.0 | 49.6 | 76.2 | 4.0 | 6.4 | 43.2 | 3.0 | 38.9 | 49.1 | 0.0 | 11.7 | 46.8 |
| | | 800 | 52.4 | 0.0 | 48.5 | 80.2 | 7.0 | 8.9 | 56.8 | 3.0 | 41.9 | 52.5 | 0.0 | 5.7 | 53.5 |
| | | 1200 | 70.5 | 0.0 | 47.4 | 88.6 | 6.0 | 8.4 | 69.1 | 2.0 | 45.2 | 60.7 | 0.0 | 5.0 | 71.2 |
| | PGD | 400 | 48.4 | 0.0 | 46.4 | 74.8 | 5.0 | 4.9 | 47.1 | 1.0 | 40.2 | 49.1 | 0.0 | 14.2 | 42.8 |
| | | 800 | 51.1 | 0.0 | 44.1 | 77.2 | 3.0 | 7.7 | 52.6 | 5.0 | 45.7 | 52.5 | 0.0 | 4.8 | 49.6 |
| | | 1200 | 67.3 | 0.0 | 45.3 | 82.6 | 8.0 | 6.2 | 57.7 | 6.0 | 47.8 | 60.7 | 0.0 | 7.6 | 66.4 |
| | Chameleon | 400 | 50.2 | 0.0 | 43.4 | 64.8 | 8.0 | 9.0 | 47.0 | 13.0 | 39.1 | 42.2 | 0.0 | 14.2 | 42.8 |
| | | 800 | 62.6 | 0.0 | 46.3 | 78.0 | 5.0 | 7.9 | 55.6 | 29.0 | 42.7 | 52.5 | 0.0 | 4.8 | 59.6 |
| | | 1200 | 71.9 | 0.0 | 43.0 | 84.6 | 5.0 | 7.2 | 67.1 | 26.0 | 44.3 | 60.7 | 0.0 | 7.6 | 66.4 |
| | DarkFed | 400 | 65.5 | 0.0 | 43.6 | 80.5 | 13.0 | 6.1 | 54.7 | 12.0 | 40.6 | 60.6 | 9.0 | 4.4 | 60.0 |
| | | 800 | 80.1 | 5.0 | 43.3 | 85.6 | 12.0 | 6.5 | 72.4 | 47.0 | 45.2 | 67.4 | 4.0 | 8.0 | 78.2 |
| | | 1200 | 88.2 | 3.0 | 44.2 | 89.2 | 15.0 | 6.9 | 78.4 | 36.0 | 43.9 | 71.4 | 7.0 | 6.8 | 82.4 |
| | A3FL | 400 | 94.5 | 0.0 | 43.5 | 100.0 | 9.0 | 8.8 | 79.5 | 7.0 | 39.9 | 89.9 | 0.0 | 4.5 | 94.6 |
| | | 800 | 96.9 | 0.0 | 42.2 | 100.0 | 7.0 | 10.5 | 77.5 | 5.0 | 41.6 | 96.7 | 0.0 | 10.7 | 96.7 |
| | | 1200 | 100 | 0.0 | 44.1 | 100.0 | 9.0 | 14.9 | 85.2 | 2.0 | 45.9 | 99.5 | 0.0 | 11.9 | 98.4 |
| | PFedBA | 400 | 90.1 | 0.0 | 44.9 | 100.0 | 10.0 | 9.3 | 74.5 | 7.0 | 32.4 | 83.2 | 0.0 | 5.0 | 88.7 |
| | | 800 | 93.5 | 0.0 | 44.3 | 100.0 | 6.0 | 11.5 | 76.1 | 3.0 | 43.9 | 92.1 | 0.0 | 9.8 | 92.9 |
| | | 1200 | 100 | 0.0 | 40.8 | 100.0 | 5.0 | 13.8 | 80.2 | 1.0 | 42.2 | 100.0 | 0.0 | 14.5 | 96.1 |
| | Mirages | 400 | 92.2 | 0.0 | 44.1 | 100.0 | 7.0 | 8.5 | 72.7 | 4.0 | 35.2 | 85.3 | 0.0 | 4.6 | 90.1 |
| | | 800 | 95.1 | 0.0 | 44.5 | 100.0 | 4.0 | 10.3 | 77.9 | 2.0 | 46.1 | 94.2 | 0.0 | 8.5 | 94.0 |
| | | 1200 | 100 | 0.0 | 40.7 | 100.0 | 5.0 | 12.5 | 82.1 | 1.0 | 48.7 | 100.0 | 0.0 | 12.9 | 98.8 |
| Tiny-ImageNet | Vanilla | 800 | 57.5 | 0.0 | 50.0 | 69.4 | 1.0 | 4.0 | 44.2 | 0.0 | 13.5 | 58.9 | 0.0 | 3.1 | 58.5 |
| | | 1200 | 70.2 | 0.0 | 49.6 | 78.0 | 4.0 | 9.0 | 65.8 | 0.0 | 12.8 | 69.9 | 0.0 | 19.4 | 70.4 |
| | | 1600 | 81.4 | 0.0 | 46.4 | 84.2 | 4.0 | 11.9 | 72.8 | 0.0 | 11.4 | 80.1 | 0.0 | 17.5 | 82.9 |
| | PGD | 800 | 54.1 | 0.0 | 49.5 | 64.8 | 5.0 | 12.9 | 41.2 | 0.0 | 13.0 | 52.1 | 0.0 | 2.9 | 51.7 |
| | | 1200 | 67.8 | 0.0 | 52.6 | 72.9 | 5.0 | 8.4 | 61.8 | 0.0 | 16.2 | 64.9 | 0.0 | 20.6 | 67.9 |
| | | 1600 | 75.9 | 0.0 | 59.8 | 81.5 | 3.0 | 12.8 | 69.1 | 0.0 | 12.1 | 78.2 | 0.0 | 15.9 | 77.9 |
| | Chameleon | 800 | 58.4 | 0.0 | 49.2 | 71.3 | 4.0 | 7.6 | 49.8 | 0.0 | 13.1 | 56.4 | 0.0 | 6.7 | 63.2 |
| | | 1200 | 72.6 | 0.0 | 51.5 | 78.9 | 4.0 | 8.5 | 67.4 | 0.0 | 12.8 | 70.4 | 0.0 | 11.4 | 75.3 |
| | | 1600 | 84.1 | 0.0 | 50.2 | 85.9 | 5.0 | 8.8 | 77.9 | 0.0 | 12.5 | 82.1 | 0.0 | 12.5 | 86.2 |
| | DarkFed | 800 | 74.2 | 0.0 | 49.8 | 78.3 | 9.0 | 9.1 | 70.6 | 20.0 | 14.2 | 68.2 | 0.0 | 4.6 | 73.1 |
| | | 1200 | 85.6 | 0.0 | 49.3 | 88.2 | 11.0 | 8.4 | 81.2 | 23.0 | 13.9 | 76.9 | 0.0 | 6.2 | 85.9 |
| | | 1600 | 90.1 | 0.0 | 50.0 | 93.4 | 10.0 | 8.8 | 88.4 | 26.0 | 13.8 | 81.4 | 0.0 | 6.8 | 91.3 |
| | A3FL | 800 | 99.5 | 0.0 | 49.0 | 100 | 0.0 | 14.0 | 99.1 | 0.0 | 12.8 | 99.8 | 0.0 | 8.9 | 99.2 |
| | | 1200 | 98.8 | 0.0 | 49.1 | 100 | 1.0 | 13.2 | 97.2 | 0.0 | 11.5 | 99.1 | 0.0 | 15.0 | 99.0 |
| | | 1600 | 100.0 | 0.0 | 50.2 | 100 | 0.0 | 12.9 | 98.9 | 0.0 | 13.9 | 99.2 | 0.0 | 17.2 | 99.8 |
| | PFedBA | 800 | 96.2 | 0.0 | 50.8 | 100 | 0.0 | 18.6 | 96.2 | 0.0 | 12.2 | 97.2 | 0.0 | 9.8 | 98.9 |
| | | 1200 | 97.6 | 0.0 | 50.1 | 100 | 2.0 | 17.5 | 95.8 | 0.0 | 12.8 | 98.3 | 0.0 | 14.2 | 98.7 |
| | | 1600 | 99.5 | 0.0 | 51.8 | 100 | 1.0 | 16.8 | 95.9 | 0.0 | 11.5 | 99.8 | 0.0 | 15.0 | 98.9 |
| | Mirages | 800 | 98.5 | 0.0 | 50.1 | 100 | 0.0 | 17.6 | 98.1 | 0.0 | 14.1 | 98.1 | 0.0 | 9.2 | 99.2 |
| | | 1200 | 98.9 | 0.0 | 50.8 | 100 | 2.0 | 18.1 | 97.3 | 0.0 | 15.2 | 99.2 | 0.0 | 13.9 | 99.8 |
| | | 1600 | 99.5 | 0.0 | 50.4 | 100 | 1.0 | 17.2 | 98.3 | 0.0 | 14.5 | 100 | 0.0 | 14.8 | 100.0 |

Table 6: Comparison of TPR, FPR and ASR under different Dirichlet settings $d$ and poisoned learning rates.

| $d$ | plr | Vanilla | | | | | PGD | | | | | A3FL | | | | |
|---|---|---|---|---|---|---|---|---|---|---|---|---|---|---|---|---|
| | | TPR | FPR | ASR | ACC | LASR | TPR | FPR | ASR | ACC | LASR | TPR | FPR | ASR | ACC | LASR |
| 0.1 | 0.01 | 94.0 | 1.8 | 8.6 | 83.5 | 63.4 | 94.0 | 2.6 | 9.3 | 83.1 | 61.1 | 100.0 | 1.3 | 14.2 | 83.4 | 100.0 |
| | 0.025 | 95.0 | 2.1 | 9.1 | 83.3 | 80.2 | 95.0 | 2.8 | 9.6 | 83.3 | 78.2 | 100.0 | 1.2 | 14.4 | 83.6 | 100.0 |
| | 0.05 | 95.0 | 2.0 | 9.5 | 83.1 | 85.6 | 96.0 | 3.1 | 9.9 | 83.3 | 84.6 | 100.0 | 1.1 | 14.6 | 83.5 | 100.0 |
| | 0.08 | 98.0 | 2.2 | 9.3 | 83.0 | 86.2 | 98.0 | 3.0 | 9.8 | 83.2 | 86.0 | 100.0 | 1.0 | 14.8 | 83.3 | 100.0 |
| 0.5 | 0.01 | 95.0 | 1.6 | 9.0 | 88.5 | 65.1 | 94.0 | 2.5 | 9.5 | 88.7 | 63.5 | 100.0 | 1.3 | 14.0 | 88.0 | 100.0 |
| | 0.025 | 95.0 | 1.7 | 9.2 | 88.1 | 81.4 | 94.0 | 2.7 | 9.6 | 88.9 | 80.1 | 100.0 | 1.2 | 14.1 | 88.2 | 100.0 |
| | 0.05 | 95.0 | 1.5 | 9.1 | 88.2 | 87.2 | 94.0 | 2.6 | 9.7 | 88.8 | 85.3 | 100.0 | 1.2 | 14.2 | 88.1 | 100.0 |
| | 0.08 | 99.0 | 1.8 | 9.3 | 88.0 | 88.1 | 96.0 | 2.8 | 9.8 | 88.9 | 87.1 | 100.0 | 1.1 | 14.3 | 88.0 | 100.0 |
| 0.9 | 0.01 | 95.0 | 1.1 | 8.8 | 92.0 | 66.2 | 96.0 | 1.9 | 8.5 | 91.9 | 64.1 | 100.0 | 1.4 | 14.8 | 91.8 | 100.0 |
| | 0.025 | 96.0 | 1.2 | 8.9 | 92.0 | 83.1 | 97.0 | 2.1 | 8.6 | 91.8 | 80.1 | 100.0 | 1.3 | 14.9 | 91.9 | 100.0 |
| | 0.05 | 96.0 | 1.0 | 8.9 | 92.1 | 88.9 | 97.0 | 2.0 | 8.4 | 92.0 | 86.8 | 100.0 | 1.5 | 15.0 | 91.9 | 100.0 |
| | 0.08 | 100.0 | 1.3 | 9.0 | 91.9 | 88.6 | 98.0 | 2.2 | 8.7 | 91.7 | 87.3 | 100.0 | 1.6 | 15.1 | 91.7 | 100.0 |

| $d$ | plr | Chameleon | | | | | DarkFed | | | | | PFedBA | | | | |
|---|---|---|---|---|---|---|---|---|---|---|---|---|---|---|---|---|
| | | TPR | FPR | ASR | ACC | LASR | TPR | FPR | ASR | ACC | LASR | TPR | FPR | ASR | ACC | LASR |
| 0.1 | 0.01 | 95.0 | 2.0 | 9.3 | 83.0 | 84.1 | 100.0 | 3.8 | 9.0 | 83.7 | 88.2 | 100.0 | 2.5 | 13.5 | 83.1 | 100.0 |
| | 0.025 | 95.0 | 2.2 | 9.4 | 83.9 | 88.0 | 100.0 | 4.1 | 9.0 | 83.5 | 93.1 | 100.0 | 2.7 | 13.7 | 83.3 | 100.0 |
| | 0.05 | 95.0 | 2.1 | 9.6 | 83.9 | 90.9 | 100.0 | 4.5 | 9.2 | 83.4 | 96.8 | 100.0 | 2.6 | 13.9 | 83.2 | 100.0 |
| | 0.08 | 97.0 | 2.3 | 9.5 | 83.8 | 91.2 | 100.0 | 4.2 | 9.1 | 83.3 | 99.4 | 100.0 | 2.4 | 14.0 | 83.0 | 100.0 |
| 0.5 | 0.01 | 95.0 | 2.0 | 9.3 | 88.8 | 84.9 | 100.0 | 3.0 | 8.8 | 88.6 | 89.1 | 100.0 | 2.1 | 14.3 | 88.7 | 100.0 |
| | 0.025 | 95.0 | 2.1 | 9.4 | 88.0 | 89.3 | 100.0 | 3.1 | 8.9 | 88.8 | 94.0 | 100.0 | 2.2 | 14.4 | 88.9 | 100.0 |
| | 0.05 | 97.0 | 2.0 | 9.4 | 88.1 | 91.2 | 100.0 | 3.2 | 8.9 | 88.9 | 97.2 | 100.0 | 2.0 | 14.5 | 88.8 | 100.0 |
| | 0.08 | 98.0 | 2.2 | 9.5 | 88.2 | 92.6 | 100.0 | 3.4 | 9.0 | 88.7 | 100.0 | 100.0 | 2.3 | 14.6 | 88.6 | 100.0 |
| 0.9 | 0.01 | 95.0 | 2.2 | 9.2 | 91.0 | 85.3 | 100.0 | 2.0 | 9.0 | 92.0 | 91.0 | 100.0 | 1.8 | 13.2 | 91.7 | 100.0 |
| | 0.025 | 95.0 | 2.4 | 9.3 | 91.2 | 90.6 | 100.0 | 2.3 | 9.1 | 92.2 | 95.9 | 100.0 | 2.0 | 13.3 | 91.8 | 100.0 |
| | 0.05 | 98.0 | 2.3 | 9.3 | 91.1 | 93.5 | 100.0 | 2.1 | 9.1 | 92.1 | 98.2 | 100.0 | 1.9 | 13.1 | 91.8 | 100.0 |
| | 0.08 | 98.0 | 2.5 | 9.4 | 91.0 | 94.1 | 100.0 | 2.5 | 9.2 | 91.9 | 100.0 | 100.0 | 2.1 | 13.4 | 91.6 | 100.0 |

neutralize backdoor effects when sufficient data is available. Considering the defender's capability, we report results based on a dataset size of 1000.

**Visualization of Activation.** Figure 11 shows the activation values on backdoor, OOD and clean samples. The results show that the activation positions of OOD samples almost completely cover

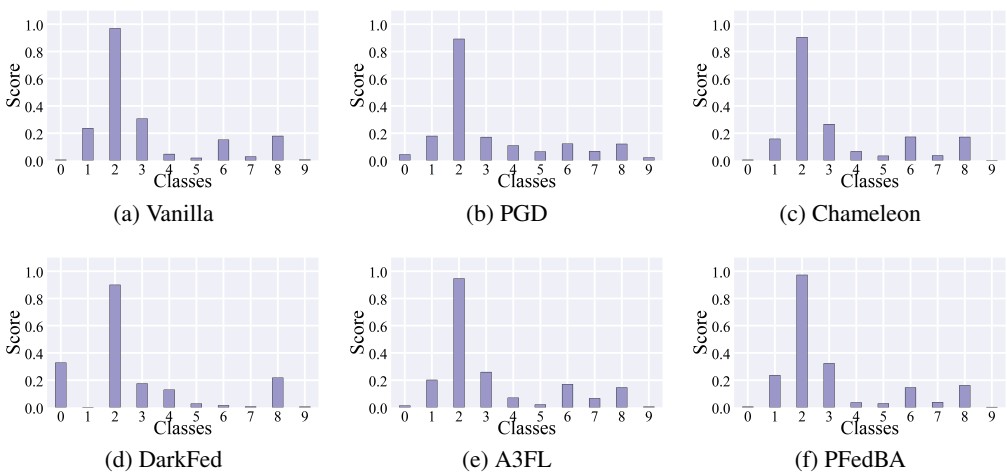

Figure 9: Visualization of *IO*Shift scores corresponding to Table 1.

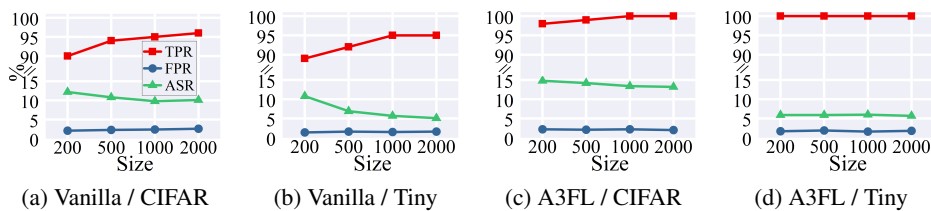

Figure 10: Performance of *IO*Shift under different sizes of ID and OOD dataset.

that of the ground truth backdoor samples, indicating that we can use OOD samples to accurately locate the neurons related to the backdoor samples.

**Different Network Architectures.** Table 7 shows the performance of *IO*Shift under different widely used network architectures. Experiments on VGG16 and ResNet50 demonstrate that *IO*Shift remains robust when the model architecture changes. This robustness comes from the fact that *IO*Shift relies on the underlying principle of backdoor-induced activation pathways, which is independent of network structures.

Table 7: Performance of *IO*Shift on different network architectures.

| Attack | VGG16 | | | ResNet50 | | |
|---|---|---|---|---|---|---|
| | TPR | FPR | ASR | TPR | FPR | ASR |
| Vanilla | 94.0 | 3.2 | 9.2 | 94.0 | 10.3 | 13.4 |
| PGD | 92.0 | 3.6 | 10.1 | 91.0 | 10.1 | 12.1 |
| Chameleon | 94.0 | 2.9 | 9.6 | 93.0 | 9.2 | 13.2 |
| DarkFed | 98.0 | 3.3 | 9.7 | 100.0 | 9.5 | 8.6 |
| A3FL | 100.0 | 3.4 | 14.1 | 96.0 | 8.9 | 15.1 |
| PFedBA | 100.0 | 3.2 | 13.7 | 95.0 | 9.2 | 14.8 |

**Removal Performance under A3FL attack.** Figure 12 shows the comparison of accuracy of global model and SOTA methods at different training epochs under A3FL attack. The results show that *IO*Shift outperforms other defenses, regardless of which training epoch.

**Computational Costs.** Table 8 shows the running time of backdoor detection within one epoch of FL under Tiny-ImageNet dataset and A3FL attack. The experiments are conducted on a system equipped with an i7-9700K CPU and a GeForce RTX 2060 Super GPU. Although our method is not the fastest in terms of running time (12 seconds), it achieves an exceptionally high TPR of 100.0% and an impressively low FPR of 1.7%, along with a low ASR of 6.5%. Compared to other methods, *IO*Shift

Table 8: Running times.

| | Flame | FDCR | Indicator | AlignIns | *IO*Shift |
|---|---|---|---|---|---|
| TPR (%) | 0.0 | 44.0 | 73.0 | 40.2 | 100.0 |
| FPR (%) | 43.2 | 37.7 | 41.2 | 37.5 | 1.7 |
| ASR (%) | 99.3 | 74.8 | 78.5 | 76.2 | 6.5 |
| Time (s) | 5.2 | 10.1 | 19.6 | 6.4 | 12.1 |

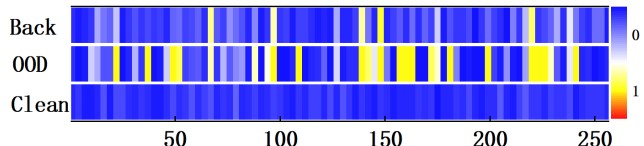

Figure 11: Visualization of activation on 50 backdoor, OOD and clean samples.

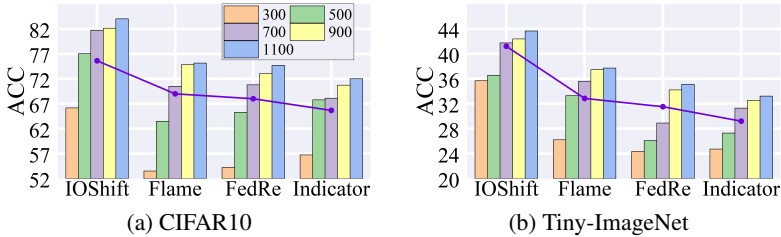

(a) CIFAR10           (b) Tiny-ImageNet

Figure 12: Comparison of removal performance on Accuracy (ACC) under different global training epochs and different defenses under A3FL attack.

demonstrates a superior balance between detection performance and robustness, making it a highly effective solution for backdoor detection. *IO*Shift consists of three major steps: Forward Inference on ID and OOD samples, IOShift Score Computation (via averaging and subtraction) and Removal under Fisher Information (on OOD data only).

Fortunately, IOShift supports scalability with respect to class number by allowing a larger batch size during inference. The detailed complexity analysis are as followed: (1) Suppose the number of classes is $N$, and we select $K_I = 20$ labeled samples per class. For OOD data, we use $M_O$ unlabeled samples. Therefore, total forward inference count is: $K_I * N + M_O$. With batch size $B$, the computational complexity is: $O((K_I * N + M_O)/B)$. (2) IOShift Score computation only involves vector addition and averaging. Its costs are negligible compared to forward inference. (3) Removal under Fisher Information computes Fisher Information on only OOD samples. Complexity is approximately: $O((M_O * F)/B)$, where $F$ is per-sample cost on Fisher information computation. This step is independent of the number of classes because it only conduct adaptive pruning for target class, whose number is usually small. When applied to larger-scale datasets, IOShift's time complexity grows linearly with the number of classes, approximately: $O((K_I * N)/B)$. However, this cost can be significantly reduced by using a reasonable batch size (e.g., $B = 256$).