# OpenReview forum: "IOShift: Backdoor Defense via Model Bias Shift in Federated Learning"
_ICLR.cc/2026/Conference — Submitted to ICLR 2026_

### Official Review · Reviewer_KWTU · 2025-10-15

**Soundness:** 2
**Presentation:** 2
**Contribution:** 2
**Rating:** 2
**Confidence:** 4

**Summary:**

This paper identifies the Backdoor-Induced Model Bias Shift phenomenon, where backdoors create a stronger malicious activation path between the trigger and target class, causing abnormal bias toward out-of-distribution (OOD) data. Based on this insight, the authors propose IOShift, a federated backdoor detection and removal framework that leverages model bias shift between in-distribution (ID) and OOD data. IOShift detects backdoors by identifying classes with significant bias shifts and removes them without requiring predefined benign parameter spaces, making it effective under highly Non-IID settings. It can be easily integrated into existing FL systems, and experiments show that IOShift achieves superior performance in both detection and mitigation compared with prior methods.

**Strengths:**

1. The method is easy to understand.
2. The authors validate the effectiveness of IOShift through extensive experiments.

**Weaknesses:**

1. Although IOShift is presented as a federated learning (FL) defense framework, its detection procedure operates on each client model independently, without leveraging any FL-specific properties such as aggregation dynamics or client interaction. Therefore, comparisons against centralized (non-federated) backdoor detection methods would be more appropriate and informative.

2. The intuition underlying IOShift appears to be unreliable. For instance, in Figures 2(a) and 2(c), the benign client’s predictions on in-distribution (ID) data are heavily biased toward label 8, which is conceptually unreasonable. This raises concerns about whether the proposed bias-shift measurement truly captures backdoor-related behavior rather than random model bias or data imbalance.

3. The key observation—“if a backdoor path exists, OOD data will also activate this shortcut path”—lacks theoretical or empirical justification. If such shortcut paths could indeed be easily triggered by OOD samples, backdoor detection would be a trivial problem, which contradicts existing research indicating its intrinsic difficulty. The authors should provide stronger evidence or theoretical reasoning to support this claim.

4. The paper claims that IOShift performs well under high Non-IID conditions, yet it does not present any experiments that explicitly analyze the impact of Non-IID degree on performance. Based on the described mechanism, IOShift may degrade significantly under low Non-IID or near-IID settings, where model bias shift becomes less distinguishable.

5. IOShift requires a relatively large amount of ID data (20 samples per class, i.e., 400 samples in total). While some prior defenses also assume access to clean data, their requirements are much lighter—for example, FLTrust only uses 10 samples. This high data dependence limits the practicality of IOShift in real-world federated deployments.

6. The experiments do not consider adaptive attackers who are aware of the IOShift defense and can modify their backdoor strategy accordingly. Evaluating IOShift under such adaptive settings is crucial to demonstrate its robustness beyond standard attacks.

7. The proposed method is only evaluated on vision datasets. It remains unclear whether the same bias-shift intuition and detection mechanism would generalize to other modalities such as NLP, where model behavior and activation dynamics differ substantially.

**Questions:**

Refer to “Weaknesses”.

---

### Official Review · Reviewer_tvPd · 2025-10-31

**Soundness:** 3
**Presentation:** 4
**Contribution:** 3
**Rating:** 4
**Confidence:** 5

**Summary:**

The proposes a new defense framework for detecting and mitigating backdoor attacks in Federated Learning (FL). The paper identifies a phenomenon termed Backdoor-Induced Model Bias Shift, where backdoor attacks cause a model’s bias on out-of-distribution (OOD) data to shift toward the target class while suppressing bias for in-distribution (ID) data. Leveraging this insight, IOSHIFTquantifies the bias shift through an IOShift score derived from ID/OOD bias statistics to detect malicious updates. To mitigate detected backdoors, IOSHIFT employs an adaptive weight pruning strategy guided by the IOShift score, selectively removing backdoor-related parameters while preserving clean model performance. Experiments on CIFAR-10 and Tiny-ImageNet against a range of backdoor attacks and defenses demonstrate that IOSHIFT achieves superior detection accuracy (high TPR, low FPR) and lower attack success rates (ASR), particularly under challenging Non-IID FL scenarios.

**Strengths:**

1. Introduces and empirically validates the *Backdoor-Induced Model Bias Shift*, offering a new conceptual understanding of backdoor behavior beyond conventional parameter-space deviations.
2. Provides an integrated detection-and-mitigation pipeline; the adaptive pruning guided by the IOShift score effectively balances backdoor removal with minimal clean accuracy degradation.
3. Demonstrates strong robustness in Non-IID FL environments, where many existing anomaly detection–based defenses fail due to the natural heterogeneity of benign updates.

**Weaknesses:**

1. The defense entirely depends on the presence of a measurable bias shift. A sophisticated adversary could design backdoors that avoid or invert this shift, thereby evading detection. Once the model bias–based criterion becomes public, adaptive attacks minimizing or mimicking benign bias patterns could undermine IOSHIFT’s core detection mechanism.
2. The detection (α) and pruning (β) thresholds are empirically chosen and static. Without an adaptive calibration strategy, IOSHIFT may exhibit high false positives or false negatives under unseen or dynamic attack settings. A principled or data-driven thresholding method would improve robustness.
3. Although adaptive pruning aims to minimize harm to clean tasks, any form of neuron pruning can reduce generalization, particularly for complex datasets or when aggressive pruning is needed. Additional quantitative analysis or guarantees on clean accuracy retention would strengthen the paper’s claims.
4. Calculating neuron importance via the Fisher Information Matrix on OOD data and performing iterative pruning increases complexity and introduces new hyperparameters. Discussion of scalability to large models or resource-constrained FL environments would improve practicality.
5. The paper states that runtime overhead is “slightly higher than gradient checking but negligible compared to client training.” Providing a small table or quantitative comparison of pruning overhead relative to one FL aggregation round would substantiate this claim.
6. Experiments are restricted to color-image datasets (CIFAR-10 and Tiny-ImageNet). Evaluating IOSHIFT on grayscale datasets such as MNIST or FEMNIST would provide additional evidence of robustness and generalizability across different data modalities.

**Questions:**

Please address the aforementioned weaknesses, particularly regarding vulnerability to adaptive adversaries, threshold adaptivity, scalability, and evaluation on grayscale datasets.

---

### Official Review · Reviewer_ANdZ · 2025-11-01

**Soundness:** 3
**Presentation:** 2
**Contribution:** 3
**Rating:** 6
**Confidence:** 5

**Summary:**

This paper proposes a defense method named IOShift to counter backdoor attacks in federated learning. The authors motivate their approach by observing that backdoored models exhibit distinct prediction discrepancies between in-distribution and out-of-distribution data. IOShift leverages this insight by examining the bias between these predictions and applying an adaptive pruning strategy to remove backdoor-related information. Extensive experiments on two vision datasets are conducted to demonstrate the effectiveness of the proposed method.

**Strengths:**

1. The paper is generally well-written and easy to follow.

2. The motivation is clearly articulated and presents a compelling case for the proposed approach.

3. The authors conduct an extensive empirical evaluation of IOShift on two vision datasets, comparing its performance against a set of representative recent defense methods.

**Weaknesses:**

1. The paper lacks a theoretical analysis of IOShift's robustness. A formal understanding of why and when IOShift is expected to succeed would strengthen the paper and provide insight into its generalizability.

2. IOShift may introduce significant computational overhead, as it requires evaluating each local model on both ID and OOD datasets to determine target classes and perform pruning. This could be particularly problematic in cross-device FL settings with a large number of clients and constrained resources.

3. The empirical evaluation is limited to vision datasets. The effectiveness and generalizability of IOShift would be better demonstrated if the authors included experiments on other data modalities, such as text or tabular data.

4. The readability of the figures and tables is not good.

**Questions:**

1. Is the averaging in Equation (2) performed over all clients?

2. How does the performance of IOShift vary with the number of OOD samples available for defense?

3. Will the authors release the code? The abstract states "Code is available here," but no hyperlink or repository link is provided.

---

### Official Review · Reviewer_FKex · 2025-11-01

**Soundness:** 2
**Presentation:** 3
**Contribution:** 2
**Rating:** 4
**Confidence:** 5

**Summary:**

This paper reveal the backdoor induced model bias shift phenomenon and propose IOShift which utilize the bias shift on OOD data as a reference for backdoor detection.

**Strengths:**

1. The concept of Backdoor-Induced Model Bias Shift is well-motivated and empirically validated, offering a fresh perspective beyond gradient-based or similarity-based defenses.
2. IOShift does not rely on defining a benign parameter space, making it robust under high data heterogeneity.

**Weaknesses:**

1. The IOShift requires a small ID dataset and unlabeled OOD data, in some extend, this assumption might not be hold for sensitive scenarios. For privacy, the ID dataset might not exsit.
2. The detection threshold $\alpha$ and removal threshold $\beta$ are hyperparameters that need careful tuning. The paper provides some guidance but does not fully address how to set them adaptively in practice.
3. The running time of IOShift is reported to be "a bit more" than simpler baseline defenses like gradient checking schemes. Although the authors argue this is negligible compared to client training time, for resource-constrained or very large-scale FL systems, this kind of increase might be still non-trivial.

**Questions:**

See weakness

---

### Meta-Review · Area_Chair_SXVZ · 2026-01-06

**Summary:**

This paper identifies a phenomenon termed Backdoor-Induced Model Bias Shift and proposes IOShift, a federated backdoor detection and mitigation framework that leverages prediction bias differences between in-distribution and out-of-distribution data. Reviewers generally found the intuition interesting and acknowledged that the method is clearly motivated and empirically evaluated on vision benchmarks. However, the suggested decision is mainly informed by substantial concerns about the reliability and generality of the bias-shift assumption, limited theoretical or mechanistic grounding, and questions about practicality and robustness in realistic federated settings.

**Reviewer Concerns:**

Several reviewers questioned whether the observed bias shift is a stable and principled indicator of backdoors, noting the lack of theoretical justification and the possibility that the measured effect may reflect spurious bias or data imbalance rather than backdoor behavior. Concerns were repeatedly raised about vulnerability to adaptive attackers, dependence on manually chosen detection and pruning thresholds, and the lack of a clear strategy for calibrating them in unseen settings. Practical limitations were also emphasized, including reliance on a non-trivial amount of clean ID data, additional computational overhead, and unclear scalability to large-scale or cross-device FL. Finally, the evaluation is restricted to vision datasets, with no analysis across modalities or systematic study of non-IID degree, leaving generalizability insufficiently supported.

**Reviewer Scores:**

No rebuttal or discussion signals were provided. The initial scores span from 2 to 6, with two reviewers clearly below threshold and others expressing significant reservations. Given that the main concerns target the core assumption, robustness, and applicability of the method rather than missing details, I believe most reviewers would likely maintain their original scores.

---

### Decision · Program_Chairs · 2026-01-26

Reject